# Synthesis and Characterization of Hybrid Fiber-Reinforced Polymer by Adding Ceramic Nanoparticles for Aeronautical Structural Applications

**DOI:** 10.3390/polym13234116

**Published:** 2021-11-26

**Authors:** Omar Talal Bafakeeh, Walid Mahmoud Shewakh, Ahmed Abu-Oqail, Walaa Abd-Elaziem, Metwally Abdel Ghafaar, Mohamed Abu-Okail

**Affiliations:** 1Industrial Engineering Department, Faculty of Engineering, Jazan University, Jazan P.O. Box 114, Saudi Arabia; Albafakeeh@gmail.com (O.T.B.); Waleedshewakh@hotmail.com (W.M.S.); 2Mechanical Production Department, Faculty of Technology Education, Beni-Suef University, Beni-Suef P.O. Box 62521, Egypt; ahmed_abuoqail@yahoo.com; 3Department of Mechanical Design and Production Engineering, Faculty of Engineering, Zagazig University, Zagazig P.O. Box 44519, Egypt; walaa.abdelaal@ejust.edu.eg; 4Manufacturing Engineering and Production Technology Department, Modern Academy for Engineering and Technology, Cairo P.O. Box 11571, Egypt; Dr.Metwally123@gmail.com

**Keywords:** aeronautical structural applications, hybrid fibers reinforced polymer, composites, mechanical response, structural behavior

## Abstract

The multiscale hybridization of ceramic nanoparticles incorporated into polymer matrices reinforced with hybrid fibers offers a new opportunity to develop high-performance, multifunctional composites, especially for applications in aeronautical structures. In this study, two different kinds of hybrid fibers were selected, woven carbon and glass fiber, while two different ceramic nanoparticles, alumina (Al_2_O_3_) and graphene nanoplatelets (GNPs), were chosen to incorporate into a polymer matrix (epoxy resin). To obtain good dispersion of additive nanoparticles within the resin matrix, the ultrasonication technique was implemented. The microstructure, XRD patterns, hardness, and tensile properties of the fabricated composites were investigated here. Microstructural characterization demonstrated a good dispersion of ceramic nanoparticles of Al_2_O_3_ and GNPs in the fabricated composites. The addition of GNPs/Al_2_O_3_ nanoparticles as additive reinforcements to the fiber-reinforced polymers (FRPs) induced a significant increase in the hardness and tensile strength. Generally, the FRPs with 3 wt.% nano-Al_2_O_3_ enhanced composites exhibit higher tensile strength as compared with all other sets of composites. Particularly, the tensile strength was improved from 133 MPa in the unreinforced specimen to 230 MPa in the reinforced specimen with 3 wt.% Al_2_O_3_. This can be attributed to the better distribution of nanoparticles in the resin polymer, which, in turn, induces proper stress transfer from the matrix to the fiber phase. The hybrid mode mechanism depends on the interaction among the mechanical properties of fiber, the physical and chemical evolution of resin, the bonding properties of the fiber/resin interface, and the service environment. Therefore, the hybrid mode of woven carbon and glass fibers at a volume fraction of 64% with additive nanoparticles of GNPs/Al_2_O_3_ within the resin was appropriate to produce aeronautical structures with extraordinary properties.

## 1. Introduction

Aeronautical structural applications have become an increasingly attractive and very important issue when it comes to the complex issue of the different loading rates [1,2,3,4,5,6,7,8,9], especially involving thin-walled structures (TWS) [10,11,12]. TWS can be produced by using fiber-reinforced polymers (FRPs) [13,14]. TWS made with FRPs have numerous advantages, such as high strength, high corrosion resistance, low weight, low cost, and ease of fabrication [15]. Owing to these numerous advantages, TWS produced by FRPs are considered suitable for aeronautical structural applications [16]. 

Particularly, various aeronautical structures, such as wings, flight controls, fuselages, stabilizers, landing gear, as well as power plants, are exposed to widely varying loads during repeated aviation operation [17]. In other words, one flight necessitates many maneuvers, which can cause different modes of loading [18]. Thus, outstanding features of durability and reliability are the main prerequisites for fabricating modern aeronautical structures [19]. To meet these requirements, there are many contradictory limitations during the design process of airplanes [18]. Some of these limitations involve: (i) the purpose of aircraft’s usage or intended purpose (passenger or fighter aircraft); (ii) aircraft regulations, which represent the conditions for acquiring certification for a new aircraft design; and (iii) financial factors such as market, environmental, and safety, etc. These restrictions make aeronautical structural applications some of the most complex products of modern technology [20]. Moreover, one of the most critical issues is the component’s mass, which has a great impact on both technical and flight characteristics, as well as cost-effectiveness [21]. However, most aeronautical structures are constructed as TWS, which perfectly fulfills the assumption of reduced component weight [22]. Sometimes, to obtain the properties of rigidity and strength in aeronautical structural applications, longitudinal and transverse elements can be used to reinforce the exterior [23]. On the other hand, in the aeronautical structural applications, a loss of stability of the exterior must be permissible where exceeding the load levels will result in reaching the critical load and causing damage to the structure. Therefore, the many functions of aeronautical structural components must be considered before the design stage in to satisfactory to all the requirements of airlines. However, some of these functions mentioned determine the importance of required properties and their relation to different loading rates, as shown in Figure 1 [24]. One of the most important aeronautical structural components is the cockpit. Other important components, among many, are the fuselage, turbine engine, slats, wings, spoiler, winglets, aileron, flaps, horizontal and vertical stabilizers, elevators, and rudder [24].

However, many components of aeronautical structures can be produced with FRPs [25]. Specifically, belly fairing skins, radomes, and pylon fairing can be fabricated by using aramid fiber-reinforced polymers (AFRPs). In contrast, nose landing-gear doors, engine cowls, main landing-gear led fairings, flaps, spoilers, floor panels, and trailing edges can be produced by using carbon fiber-reinforced polymers (CFRPs). Fin boxes, rudders, leading edges, fin/fuselage fairing, and trailing-edge panels can be fabricated by using glass fiber-reinforced polymers (GFRPs). Nevertheless, aeronautical structures that are produced by using FRPs face complex and varied loading rates. Some of the challenges associated with these loads are aerodynamic or inertial, related to the mass or density of the components, as well as environmental effects such as ultraviolet degradation and temperature change [26]. Such loads accelerate the damage of aeronautical structural components. Moreover, aeronautical structures that are made using FRPs have a varied composition of fiber and polymer in order to obtain a combination of different properties. These variations cause some limitations, such as relatively low damage-bearing capacity, anisotropic mechanical properties, and difficulty in recycling. Therefore, innovative construction methods for aeronautical structures with new engineering materials would provide many advantages. Some of these benefits are related to affordability and the reduction of varied loading rates. These advantages can be obtained by unique microstructural characteristics and mechanical properties, which make it a viable and plentiful alternative for aeronautical structural applications. To achieve this, a new, innovative system of aeronautical structural design using advanced materials must be developed.

Carbon fiber has a lot of potential as a replacement for fiberglass. It exhibits much higher stiffness and lower density than fiberglass, allowing aeronautical structures to be thinner, stiffer, and lighter. However, carbon fiber has a relatively low damage-bearing capacity, compressive strength, and ultimate strain and is much more expensive than E-glass fiber [27]. Moreover, such glass fiber- or carbon fiber-reinforced composites may display anisotropic properties, i.e., exceptional mechanical properties in the direction aligned with the matrix, but properties in transverse directions may not always be appropriate [28]. Currently, the proposed hybrid fiber-reinforced polymer (HFRP) composites have many benefits. Some of these advantages include reduced material cost, as well as improved mechanical properties, corrosion resistance, fracture toughness, and resistance to environmental effects [29,30,31]. However, materials must be designed and optimized to be much stiffer, more lightweight, and have better fatigue resistance than those currently used in order to extend the working life of aeronautical structures [32]. Achieving higher performance, efficiency, and reliability with lower costs for aeronautical structures is a difficult request. Therefore, a newly developed composite material characterized by lightweight design, fatigue resistance, high toughness, high durability, and damage tolerance, as well as being environmentally friendly and recyclable, is crucially needed. 

Nanotechnology has the potential to produce new polymer composites, especially based on graphene nanoplatelets (GNPs) or carbon nanotubes (CNTs) [33]. The use of graphene nanoplatelets (GNPs) or CNT composites in future aeronautical structures will provide several benefits [33]. One of these benefits is low rotational inertia, which causes aeronautical structures to accelerate rapidly [26]. As a result, the tip speed ratio is kept as close to constant as possible. This phenomenon allows aeronautical structures to improve their properties. On the other hand, since alumina (Al_2_O_3_) has a low production cost and outstanding mechanical properties, such as high hardness, wear resistance, low density, high melting point, and chemical-thermal stability, it is considered an ideal reinforcement of polymer matrix composites (PMCs) [34,35]. Because of these numerous advantages, polymer reinforced with GNPs and Al_2_O_3_ nanoparticles can serve as a potential composite material for aeronautical structures. Several factors influence the optimization of the nanomaterial mechanism of PMCs, especially the physical and chemical interaction between nanoparticles and epoxy resin. Some of these factors related to the physical and chemical properties are concentration and orientation, as well as dispersion method, treatment state, and manufacturing method. All these factors are related to the fibers, matrices, and fillers. Moreover, the effect of nanomaterials on the improvement mechanism of polymer depends on many factors, such as the interaction among the mechanical properties of fibers and nanoparticles, the physical and chemical evolution of resin, the bonding properties of fiber/resin, as well as nanoparticle interface and the service environment [36]. Thus, the current research will primarily focus on the production of aeronautical structural material reinforced with GNPs and Al_2_O_3_ nanoparticles, which offer lighter weight and higher strength than those produced with fiber-reinforced polymers (FRPs) only.

Some studies are currently being conducted on construction methods to improve the performance of aeronautical structural applications. Seretis et al. [36] studied the influence of reinforcing PMCs with stainless steel flakes (SSFs) on microstructure characteristics and mechanical properties. They used high-density polyethylene (HDPE) as matrices and different kinds of epoxy resins, such as low-viscosity of Araldite GY 783, and phenol-free Aradur 2965. On the other hand, they selected SSFs as reinforcement particulate and added different weights of SSFs, starting from 10%. In this study, two main categories were compared (SSFs with epoxy and SSFs with HDPE). According to observations of microstructure by scanning electron microscope (SEM) and measures of mechanical performance, such as tensile, compression and flexural strength of the tested samples, they concluded that the two categories were negatively affected. In contrast to plastic, deformation under tension was enhanced in the case of HDPE with low SSF content. Hao et al. [37] fabricated fiber-metal laminates (FMLs) using the technique of diffusion bonding. They used raw materials such as a magnesium alloy layer and continuous carbon fiber and Zn-Al alloy composite layers. The FMLs were composed of layers of magnesium alloy and a layer of continuous carbon-fiber Zn-Al composite. They elucidated that the Zn-Al alloy was tightly infiltrated and bonded with the carbon fiber and magnesium alloy layer. Through microstructural observation, it was shown that metal sheets can be bonded to carbon fiber to produce metallurgically bonded FMLs, provided the sheets have a low melting point. They also concluded that the mechanical results, especially those concerning tensile strength and elastic modulus, were improved by 103% and 41%, respectively, when compared to the original sheets of magnesium alloy. Quan et al. [38] used ductile steel fibers with high strength to improve the fracture toughness of adhesive joints. Two main categories of nano-toughened adhesives with unique fracture properties were selected to join aerospace-grade composite substrates. Steel fibers were longitudinally and transversely integrated into the adhesive layer to prevent the growth cracks. They concluded that strengthening the adhesive layer with steel fibers improved the fracture toughness in modes I and II. The additive steel fibers in the adhesive layer helped to reinforce weak areas, acting as a caulk to prevent crack growth. Loos. et al. [39] demonstrated that epoxy composites containing a small number of ceramic nanoparticles increase the lifespan of the product. Therefore, GNPs and Al_2_O_3_ were selected in this study to fabricate formulations of composite materials with extraordinary and unique mechanical properties [40,41,42].

The strengthening of hybrid carbon and glass fiber-reinforced polymers with two durable nanoceramic particles has rarely been studied [43,44]. Therefore, the main scope of the present article is the production of aeronautical structural materials reinforced with GNPs and Al_2_O_3_ nanoparticles that are much lighter weight and stronger than those produced by fiber-reinforced polymers (FRPs) only. However, in the current article, an attempt was made for the first time to produce thin-walled structures reinforced with GNPs and Al_2_O_3_ nanoparticles for aeronautical purposes by scattering the nanoparticles by high-frequency sonication technique and then using hand lay-up and compression molding techniques. Therefore, the main objective of this study is the synthesis and characterization of hybrid fiber-reinforced polymers for aeronautical structural purposes by the addition of ceramic nanoparticles. The specific objective is to improve the mechanical and microstructural properties of hybrid FRPs by adding two durable reinforcement nanoceramic particles, such as Al_2_O_3_ and GNPs, to the matrix. The experimental work, followed by the results and discussion, will be briefly discussed in Section 2 and Section 3. Our conclusion will be summarized in Section 4.

## 2. Materials and Methods

This section introduces the experimental procedures, including the selected material and its specifications, as well as the methodology and tools used. The detailed analysis methodology is illustrated in Figure 2. As is shown, the experimental procedures of the current work were clearly divided into four distinct steps. The first step is to prepare the material and treat the dispersion. The second step involves the synthesis and processing of the produced hybrid carbon and glass fiber-reinforced polymer with nanoceramic particles (GNPs and Al_2_O_3_) through compression mold technique, and then subsequent post-heating by heating is applied. The third step includes the microstructural observations. Finally, the fourth step comprises mechanical testing.

Different raw materials are used to construct different aeronautical structures in this study, such as woven E-glass fiber with 2.56 g/cm^3^ density, woven carbon fibers with 1.6 g/cm^3^ density, and Sikadur epoxy resin matrix (Sikadur 330) used to bond the layers of fiber fabric (see Figure 3). The woven carbon fibers and glass fibers were supplied by Arab World for Financial Investments Company, Cairo, Egypt. Sikadur 330 epoxy resin was purchased from Sika Corporation. On the other hand, two kinds of ceramic nanoparticles, such as aluminum oxide (Al_2_O_3_) and graphene nanoplatelets (GNPs), were used as strengthening nanoparticles in the resin matrix. The Al_2_O_3_ and GNPs were purchased from Nano Gate Company for Nanomaterials and Chemicals, Cairo, Egypt. Al_2_O_3_ nanopowders are white in color, with spherical shape (diameter ≤25nm), see Figure 3. The density of Al_2_O_3_ was 3.78 g/cm^3^, while the purity of Al_2_O_3_ was ~95%. On the other hand, the GNPs are gray in color with a fine-flake shape and an approximate size of ≤100 nm; see Figure 3. The physical and mechanical properties of applied raw materials are listed in Table 1.

After specifying and preparing the raw materials, we can introduce the details of the synthesis of the hybrid composite structures. In order to produce different kinds of composite structures, two manufacturing methods are used, such as hand lay-up and compression-molding techniques. To investigate the effect of adding ceramic nanoparticles on the microstructural and mechanical characteristics of hybrid carbon and glass fiber-reinforced polymers, three different kinds of aeronautical structures are selected in this study: (i) hybrid glass and carbon fibers without ceramic nanoparticles, (ii) hybrid glass and carbon fibers with equal 1.5 wt.% ratio Al_2_O_3_ and GNPs ceramic nanoparticles, and (iii) hybrid glass and carbon fibers with 3 wt.% Al_2_O_3_; see Table 2. Accordingly, the main parameter of this work is to determine the optimum weight percentage of ceramic nanoparticles within hybrid glass and carbon fibers for aeronautical purposes. Specifically, to construct the components of aeronautical structures by using hybrid FRPs with reinforcing ceramic nanoparticles, two techniques are implemented in this work. The structures in this work were built with four layers, consisting of two layers of glass fiber and two layers of carbon fiber.

The first technique was hand lay-up, and the second was the compression-molding technique. The hand lay-up technique was implemented first to construct part of the aeronautical structures. The first step of the hand lay-up technique was to place the woven fiber fabrics inside the smooth surface of the mold with an insulator in between plastic sheets as a release agent. After the mixture solution for the various composites was poured into the molds, it was covered with plastic transparent sheet as a release agent. At this time, the ceramic nanoparticles were prepared by treating the surface with stearic acid as a non-reactive modifier. This step is to assist with good adhesion between ceramic nanoparticles and epoxy resin. The stearic acid was added to the solution of ethyl-acetate with stirring for half hour by an electric mixer at a velocity of 900 rpm. Then, the ceramic nanoparticles were added to the mixture with stirring for another half hour. Then, the nanoparticles were washed in a solution of ethyl-acetate and filtered until the excess stearic acid was removed. After that, the nanoparticles were carefully added to the epoxy with intermittent stirring for 30 min at 500 rpm. Next, GNPs and Al_2_O_3_ nanoparticles with various weight percentages were added to Sikadur 330 epoxy resin and dispersed via simultaneous sonication and magnetic stirring, as seen in Figure 3. The dispersing process was carried out by sonication process of 0.5 cycles per second with 70% amplitude for 3 h. The operation specifications of the Hielscher ultrasonic processor were UP 200 S with 200 W and frequency of 24 kHz. Suspensions of the GNPs and Al_2_O_3_ in the Sikadur 330 epoxy resin were prepared with the addition of dispersing agents, as shown in Figure 3. The concentration of GNPs and Al_2_O_3_ in the Sikadur 330 epoxy resin was varied according to Table 2. The ratio of hardener to epoxy resin was 1:2. Following, the dispersion process, the suspensions were exposed to centrifugation at 150 g for 3 hr. Then, the mixture was added to the first layer of woven fabrics, making sure that the first layer of woven fabrics was saturated with epoxy. After that, the second layer of woven fabrics was added sequentially. Then, the process continued until the completion of the number of layers required. The curing cycle was done at 55 °C for 30 min during saturation. After that, the second technique (compression mold technique) was implemented by preparing the mold with a flat and good surface finishing, as well as covering its internal dimensions (300×200 mm) with a transparent plastic sheet as a release agent, as shown in Figure 3. The fiber volume fraction (νf) was measured experimentally according to ASTM D-3171-99. The average volume fraction was about at 64%. After finishing the fabrication stages (curing stage), the specimens were cut according to standard specifications for each test.

It is important to note that the structure of fiber layer, matrix, and fillers (nanoparticles) plays a critical role in dispersion, distribution, adhesion. Specifically, the atomic structure of carbon fiber and glass fiber is like that of graphite, which is composed of flat sheets of carbon atoms (graphene) placed in a regular hexagonal pattern. The difference between them is the way that the sheets are linked. The intermolecular strength between each sheet is relatively less (Van Der Waals), giving graphite its soft and brittle properties. On the other hand, the structure of Al_2_O_3_ nanoparticles is a fine, spherical particle shape, while the structure of GNPs is a fine flake shape. Therefore, the differences between structure shapes of these additive materials provide more benefits. Some of these benefits are: (i) helping to combine the matrix materials with fiber fabrics, (ii) reinforcing the constituents together, and (iii) eliminating any bubbles, voids, and porosities in the microstructure. In other words, the combination of the fiber fabrics, matrix, and nanoparticles has a significant benefit for developing unique mechanical, chemical, and physical properties.

After presenting the details about the synthesis of the hybrid composite structures, we can introduce the testing methods. In order to ensure good scattering within the hybrid fiber-reinforced polymer matrix composites, the test methodology in this work is focused on two main analyses (microstructural and mechanical). The microstructural observations were analyzed through optical microscopy (OM), scanning electron microscope (SEM), and X-ray diffraction (XRD), as well as energy dispersive X-ray spectroscopy (EDS). Mechanical testing was evaluated by using tensile and hardness tests, as shown in Figure 4.

The composite specimens were cut in both longitudinal and transverse directions to ensure the isotropy for the fabricated composite. In order to prepare the morphology of the fabricated composite specimens, the specimens were mounted, then mechanically ground, and polished according to the standard metallography practices. The phases present in the fabricated composites were identified by X-ray diffraction (XRD) technique using Cu K-alpha radiation (λ = 1.541 Å). XRD scans were carried out with a step size of 0.02° and a long scan range (2θ) of 5–60°. A tensile test according to ASTM standard was performed at room temperature with a strain rate of 5 mm/min. The tensile test specimens were cut into 250 mm long, 25 mm wide, and 10 mm thick strips, as shown in Figure 4. A hardness test was also carried out at room temperature using Shimadzu Vickers microhardness testing by applying a 200 g load and 10 sec dwell time on the fabricated nanocomposite samples. A total of 36 specimens was evaluated in this work, where three samples in the one case were evaluated in all tests, and then the average value was introduced. After introducing most details about the experimental procedures, we can present the results.

## 3. Results

### 3.1. Microstructral Observations

A representative optical image of the fabricated composite specimens is shown in Figure 5. Figure 5a shows numerous spherical bubbles distributed on the surface of the hybrid glass and carbon fiber sample (S_1_). As a result, additional nanoparticles between the hybrid glass fiber and carbon fiber are essential to be incorporated into the composite structure to eliminate the formation of such bubbles, as shown in Figure 5b,c. Generally, the surface area for a given volume is an important factor for nanoparticles. In other words, the geometrical shape of nanoparticles and their respective surface-area-to-volume ratios are of more importance. This factor varies with the geometrical shape of nanoparticles, i.e., particle, layered, and fibrous materials. In the current study, we used constituents of three geometric shapes; the nanoparticles were particulate, while the fiber fabric were fibers and layered materials.

Additionally, Figure 5b show the agglomeration of GNPs/Al_2_O_3_ nanoparticles, shown by yellow arrows, in the fabricated specimens (S_2_). Figure 6 shows the scanning electron microscopy on the top surface of fabricated specimens. According to Figure 6a, the sample (S_1_) indicates that there are no ceramic nanoparticles inside the woven hybrid glass and carbon fiber, which contributes to lowering the modulus and strength of the S_1_ sample. Additionally, it can be observed that the sample of S_1_ showed uneven and rough surfaces, with some embedded debris formed on its surface. On the other hand, the second and third conditions, (S_2_) and (S_3_) in Figure 5b,c, demonstrate that a smooth surface was obtained by the addition of the ceramic nanoparticles. The morphology of polymer composites can be divided into three categories, namely phase-separated (microcomposite), intercalated nanocomposites, and exfoliated nanocomposites. In the phase-separated (microcomposite), the clay nanoplatelets keep their crystal structure and the particle is in the microscale. In an intercalated (nanocomposite) stage, few polymer molecules penetrate into the fiber layers, with fixed interlayer spacing. In the exfoliated (nanocomposite) stage, the nanolayers are delaminated and individually dispersed in the continuous polymer matrix. The most desired structure for a nanoplatelet/polymer nanocomposite is for the nanofiller to be in the exfoliated state, as this provides maximum interfacial contact and best dispersion, resulting in optimum nanocomposite performance [45,46,47].

Figure 7 shows the microstructure at cross-surface of fabricated composites (S_1_, S_2_, and S_3_). It can be observed that some pores, voids, and nanobubbles between hybrid glass and carbon fibers were detected in S_1_ sample. For the samples S_2_ (Figure 7b) and S_3_ (Figure 7c), the pores and voids were decreased. However, the sample (S_3_) in Figure 7c exhibits better dispersing of Al_2_O_3_ nanoparticles. The carbon atoms are bonded together in microscopic crystals that are mostly aligned parallel to the long axis of the fiber. This alignment causes a high tensile property for the fiber. Fillers are used in polymers for a variety of reasons, namely, to reduce cost, improve processing, as well as control density, thermal conductivity, thermal expansion, electrical properties, magnetic properties, flame retardance, and to improve mechanical properties. Each filler type has different properties, depending on particle size, shape, and surface chemistry. In general, fillers can change the performance of polymer composites by changing the color, viscosity, barrier properties, curing rate, electrical and thermal properties, surface finish, shrinkage, etc. In general, the unique combination of the nanomaterial characteristics necessary to modify the polymer matrix properties, such as size, mechanical properties, and low concentration, has generated much interest in the field of nanocomposites. Synthesis properties of nanomaterial depend on several factors, such as type of nanoparticle, surface treatments, polymer matrix, synthesis methods, and polymer nanocomposites morphology.

Figure 8 shows the EDX analysis by FE-SEM of the prepared hybrid glass and carbon fibers with 1.5 wt.% GNPs and 1.5 wt.% Al_2_O_3_. A homogenous distribution of Al_2_O_3-_reinforcing phase can be observed in the resin matrix. Moreover, since the only elements observed are Al, O, and C, of which the elements O and Al represent the Al_2_O_3_ phase and the carbon element signifies the GNPs and woven fabric, this figure presents evidence of good adhesion between nanoparticles and the resin matrix via sonication process.

Figure 9 shows XRD patterns of the fabricated specimens, S_1_, S_2_, S_3_. Generally, the geometrical shape of nanoparticles and their respective surface-area-to-volume ratios are of more importance. This factor varies with the geometrical shape of nanoparticles, i.e., particle, layered, and fibrous materials. As shown in the figure, a more pronounced amorphous phase can be observed between the 2θ angles of 10°–30°. The C element was found in every case. This study observed two main broad peaks at 2θ values of 17° and 25°. These peaks at 25° correspond to the carbon phase. Specifically, the C element was observed in all cases. This is due to the similarity between carbon and glass fibers, as well as ceramic nanoparticles of GNPs. This also ensured the appearance of glass and carbon fibers in every cases. According to Figure 9, it can be noted that the intensity of diffraction peaks of Al_2_O_3_ nanoparticles in polymer matrix composites decreases, while the width of the peaks increases with an increase in the weight percentage of Al_2_O_3_ nanoparticles. This is because the size of Al_2_O_3_ nanoparticles was smaller than that of GNPs. On the other hand, the intensity of the diffraction peaks of carbon elements in polymer matrix composites increases, while the width of the peaks decreases with a decrease in the weight percentage of GNPs. This is because the nanoparticles of GNPs are agglomerated after the curing operation.

### 3.2. Mechanical Properties

In this section, we will elucidate the influence of the incorporation of ceramic nanoparticles with hybrid FRPs on mechanical properties such as microhardness and tensile tests.

Figure 10 demonstrates the microhardness analyses of the fabricated samples. The Vickers microhardness maps were distributed along the two axes (x and y) of the cross-sections to determine the influence of the addition of nanoparticles to hybrid FRPs. The measurement of hardness values for the samples S_1_, S_2_, and S_3_ are observed as of 0.6 VHN, 9 VHN, and 0.5 VHN for the minimum value and 1.8 VHN, 14 VHN, and 3 VHN for the maximum value of the measurements, respectively. It can be observed that the hardness increased with the addition of hard ceramic GNPs or Al_2_O_3_ particles in the samples S_2_ and S_3_, as compared to the S_1_ sample.

It can also be seen that S_2_, with 1.5 wt.% GNPs and 1.5 wt.% Al_2_O_3_ nano particles, has a higher hardness than sample S_3_, with 3 wt.% Al_2_O_3_. On the other hand, the highest value of hardness measurements was observed in the second condition (S_2_) in (b).

In the next section, the influence of incorporating ceramic nanoparticles in hybrid glass and carbon fibers on the strength and strain of fabricated composite is investigated. The tensile strength and strain of fabricated specimens for S_1_, S_2_, and S_3_ are shown in Figure 11. Generally, the tensile strength of the fiber-reinforced composites mainly relies on various factors, namely fiber orientation, length of fiber, strength, fiber content, fillers, bonding between fiber and matrix, and weave style [48,49]. It is clear that the tensile strength of the samples is improved by adding the ceramic particles to the resin matrix. The tensile strength values for samples S_1_, S_2_, and S_3_ are 133 MPa, 162 MPa, and 230 MPa, respectively, and the corresponding strains are 0.073, 0.049, and 0.057, respectively. Sample S_1_ has the highest strain value among the other samples, S_2_ and S_3_. In addition, it can be observed that the tensile strength of e sample S_2_, with 1.5 wt.% GNPs and 1.5 wt.% Al_2_O_3_ nanoparticles is lower than sample S_3_, with 3 wt.% Al_2_O_3_.

Based on these results, it can therefore be concluded that the addition of nanoparticles of Al_2_O_3_ as reinforcement in polymer matrix holds great promise for improved mechanical properties of the synthesized composites, especially in the case of aeronautical structural applications. For more clarification of the tensile results, the fracture morphologies of fabricated fibrous composites are demonstrated in Figure 12. According to Figure 12a, the fracture behavior of hybrid glass and carbon fibers without the addition of any nanoparticles (S_1_) displays both fiber pullout and breakage at multiple levels. The pulled-out fibers with fracture-like failed surface demonstrate a weak interfacial bonding between the matrix and fiber. For composite S_2_ with (GNPs–Al_2_O_3_) particles displayed in Figure 12b, many fiber pull-outs were detected on the fracture surface of the composites. However, for sample S_3_ with 3 wt.% Al_2_O_3_, some resin remained on the exposed fiber surfaces. After presenting the results related to the synthesis and characterization of hybrid fiber-reinforced polymer by adding ceramic nanoparticles, we can introduce the discussion.

## 4. Discussion

### 4.1. Microstructral Investigations

In this section, we will discuss the influence of the incorporation of ceramic nanoparticles with hybrid FRPs on the microstructural characterizations via optical microscope and scanning electron microscope analyses.

Figure 5 shows optical images of the fabricated composite specimens. According to Figure 4, the appearance of the bubbles in Figure 5a is owing to the use of two different types of woven fibers (glass fibers and carbon fibers), i.e., the differences between interactions, dimensions, densities, and volume fractions of the two woven fibers. These factors contributed to the formation of bubbles. With the addition of the ceramic particles (GNPs, and Al_2_O_3_) to glass fiber and carbon fiber, the spherical bubbles disappeared, as seen in Figure 5b,c. The formation of the agglomerates, as shown in Figure 5b, in the hybrid composite (S_2_) was unavoidable on the surface of hybrid glass and carbon fibers.

On the other hand, Figure 6 elucidates the scanning electron microscopy on the top surface of the fabricated specimens. According to Figure 6a, the sample (S_1_) doess not indicate ceramic nanoparticles within the woven hybrid glass and carbon fiber, which causes lower modulus and strength of the S_1_ sample. Meanwhile, it is observed that there is a good dispersion of ceramic nanoparticles of Al_2_O_3_ and GNP in the S_2_ sample and Al_2_O_3_ in the S_3_ sample on the matrix surface. This is due to the use of a high-frequency sonication technique, which aided in the proper mixing of ceramic nanoparticles within the epoxy matrix. It has been stated that the homogeneous dispersion of reinforcements in the matrix composite ensures isotropic mechanical properties and uniform stress distribution [31].

Figure 7 presents the microstructure at a cross-surface of the fabricated composites (S_1_, S_2_, and S_3_). According to Figure 7, in (a), there are some pores, voids, and nanobubbles that were detected between hybrid glass and carbon fibers without the addition of ceramic nanoparticles. Several factors, as mentioned before, such as the differences between dimensions, densities, and distances between the two different types of woven fibers, affect microstructure at the cross-surface of the fabricated composites. In samples S_2_ (Figure 7b) and S_3_ (Figure 7c), these pores and voids were decreased, which means a good mixing was achieved during the polymer composite fabrication. This can be ascribed to uniform dispersion of the nanoparticles on the entire surface and the reduction in concentration ratio, which aids in eliminating the agglomeration. Consequently, improvement in the resin interlaminar between both resin and fibers via incorporating nanoparticles wasachieved in the S_2_ sample. Moreover, these ceramic particles could transfer the concentration of the stress from resin matrix to the fiber and eventually improve the mechanical properties. It is worth noting that the layers of GNPs do not operate in the strengthening mechanism because the layers of GNPs are self-lubricating. However, the incorporation of two ceramic nanoparticles of Al_2_O_3_ and GNP helped to strengthen the adhesion between hybrid glass and carbon fibers within the resin matrix and improve the bond between the woven fabrics and the resin matrix. However, the sample (S_3_) in Figure 7c exhibits better dispersion of Al_2_O_3_ nanoparticles, resulting in a strong bond between the resin matrix and woven fabric, as can be seen in the magnified yellow rectangle. Nevertheless, the best scatter spots are detected in most positions throughout the matrix (see Figure 7d).

Figure 8 elucidates the EDX analysis by FE-SEM of the prepared hybrid glass and carbon fibers with 1.5 wt.% GNPs and 1.5 wt.% Al_2_O_3_. Generally, the synthesis properties of nanomaterials depend on several factors, such as type of nanoparticle, surface treatment, the polymer matrix, synthesis methods, and polymer nanocomposite morphology. According to Figure 8, homogenous scattering of the Al_2_O_3_-reinforcing phase in the resin matrix was observed. The homogenous distribution of Al_2_O_3_ via sonication process causes good adhesion between nanoparticles and the resin matrix.

Figure 9 elucidates XRD patterns of the fabricated specimens, S_1_, S_2_, S_3_. According to Figure 9, a more pronounced amorphous phase can be observed between the 2θ angles of 10°–30°. This phase with relatively higher-intensity peaks is attributed to the presence of amorphous carbon within the composite sample, especially for S_1_ and S_3_. The C element was found in every case, which is due to the similarities between additive nanoparticles, such as GNPs and glass and carbon fiber. The presence of weak peaks of the Al_2_O_3_ phase is due to its low weight percentage in the resin matrix. The main reason for the appearance of these elements is the similarities between the nanoparticles and woven fabrics. Specifically, the C element was observed in all cases, which is due to the resemblance of additive nanoparticles such as GNPs and glass and carbon fiber.

### 4.2. Mechanical Response

In this section, we will discuss the influence of the incorporation of ceramic nanoparticles with hybrid FRPs on mechanical properties, such as microhardness and tensile tests.

Figure 10 elucidates the microhardness analyses of the fabricated samples. According to Figure 10, it can be observed that the hardness increased with the addition of hard ceramic GNPs or Al_2_O_3_ particles to samples S_2_ and S_3_, as compared to sample S_1_. This increase is ascribed to the hard nature of the added ceramic materials and the nanoparticles that are distributed all over the resin matrix, as well as the uniform distribution of Al_2_O_3_ and GNPs nanoparticles in the nanocomposites. It can also be seen that S_2_, with 1.5 wt.% GNPs and 1.5 wt.% Al_2_O_3_ nanoparticles has a higher hardness than sample S_3_, with 3 wt.% Al_2_O_3_ because GNPs have a higher hardness value than Al_2_O_3_ nanoparticles. Finally, the highest value of hardness measurements was elucidated in the second condition (S_2_) in (b).

On the other hand, Figure 11 shows the tensile strength and strain of fabricated specimens for S_1_, S_2_, and S_3_. The values of tensile strength for the samples, S_1_, S_2_, and S_3_, are 133 MPa, 162 MPa, and 230 MPa, respectively, and the corresponding strains are 0.073, 0.049, and 0.057, respectively. Thus, the maximum enhancement of ~73% in the tensile strength was attained for S_3_ with 3 wt.% Al_2_O_3_ composite as compared with sample S_1_, which has no reinforcements. This is owing to the good dispersion of ceramic nanoparticles within the matrix of hybrid glass and carbon fibers. Therfore, by adding nanoparticles of alumina (Al_2_O_3_) and graphene nanoplatelets (GNPs) to the resin matrix, the tensile strength of the samples is improved since the nanoparticles act as additional reinforcement within the hybrid fabric fibers, which enhances the mechanical properties of the polymer matrix composite (resin) [50]. On the other hand, the highest strain value among the other samples, S_2_, and S_3_, is sample S_1_ because of the absence of ceramic particles, as well as the formed bubbles on the whole surface of S_1_, which cause an increase in its strain value. In addition, the tensile strength of sample S_2_ with 1.5 wt.% GNPs and 1.5 wt.% Al_2_O_3_ nanoparticles was observed to be lower than that of sample S_3_ with 3 wt.% Al_2_O_3_. This can be attributed to the agglomeration of the reinforcements (GNPs and Al_2_O_3_) in the resin matrix (see in Figure 5b), which induces the stress concentration inside the composite and consequently impairs the mechanical properties of the fabricated composite. This implies that this condition is largely responsible for lowering the mechanical properties of the polymer reinforced with ceramics [33]. Consequently, the combination of the fiber fabrics, matrix, and nanoparticles has a significant benefit for developing unique mechanical properties.

Figure 12 shows the fracture morphologies of the fabricated fibrous composites. According to Figure 12, the fracture morphologies of the fabricated samples in (a), (b), and (c) were had pullouts and breakage at multiple levels, and two modes of breakage and shearing, respectively. It can be concluded that the fibers are more prone to breakage than to be mixed-fiber breakage, such as breakage and shearing, revealing the ability of the matrix to transfer loads effectively. However, the fracture length of the fractured fiber was at the millimeter level because the interfacial bonding strength was relatively weak, thus preventing the composite material from maximizing toughness during fracture. After presenting the whole results and discussion in a detailed and clear manner, we can present the conclusion and summary of this study in the next section.

## 5. Conclusions

In this work, we successfully synthesized and characterized hybrid fiber-reinforced polymer for aeronautical structural purposes by adding ceramic nanoparticles. Specifically, the effect of hybrids of GNPs/Al_2_O_3_ nanoparticles on the microstructure and mechanical behavior of hybrid glass and carbon fiber-reinforced epoxy composites was studied. Different parameters were chosen to produce successful thin-walled structures with extraordinary properties for the aeronautical, aerospace, and airplane industries. The parameters were different weights of ceramic nanoparticles of GNPs/ Al_2_O_3_ embedded with the hybrid glass and carbon fiber-reinforced epoxy. Various analyses, such as microstructural, morphological, and mechanical tests, were used to study the influence of embedded ceramic nanoparticles on the hybrid glass and carbon fiber-reinforced epoxy. According to the reported observations from this work, the following can be concluded:Uniform distribution of GNPs and Al_2_O_3_ with weight fraction of 1.5% of both nanoparticles was achieved in the hybrid FRPs matrix, and the spherical bubbles disappeared with the addition of ceramic nanoparticles to glass fiber and carbon fiber.According to XRD results, the diffraction patterns represent two peaks, i.e., higher-intensity peaks of amorphous carbon, due to the similarities between additive nanoparticles such as GNPs and glass and carbon fiber in all the composite samples, and weak peaks of the Al_2_O_3_ phase in samples S_2_ and S_3_ due to its low weight percentage in the resin matrix.The addition reinforcements of GNPs and Al_2_O_3_ with weight fractions of both nanoparticles at 1.5% to the resin matrix affords significant improvements in the hardness, which can be ascribed to the hard nature of these ceramic nanoparticles, which displayed a uniform distribution within the resin matrix.Additionally, the tensile properties were improved with the addition of GNPs and Al_2_O_3_ particles. The highest value of tensile strength (230 MPa) belonged to the hybrid glass and carbon fibers with 3 wt.% Al_2_O_3_, while the strength value of composite S_1_ with no reinforcements was 133 MPa.The type of layer structure of fibers, matrix, and fillers (nanoparticles) plays a critical role in scattering, distribution, and also adhesion.Finally, according to the previous findings and conclusions, it can be concluded that the properties of the combination of nanomaterials depend on several factors, such as type of nanoparticles, surface treatments, polymer matrix, synthesis methods, and polymer nanocomposite morphology. These factors will a research area of interest in the future.

These conclusions and findings can be of assistance to produce successful thin-walled structures with extraordinary properties, such as high strength-to-weight ratio, resistance to fracture toughness, fatigue, and damage

## Figures and Tables

**Figure 1 polymers-13-04116-f001:**
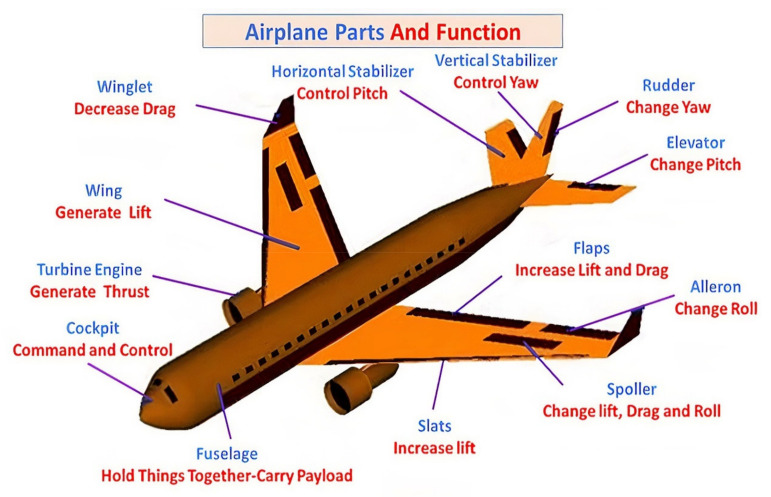
Airplane components and functions [24,26].

**Figure 2 polymers-13-04116-f002:**
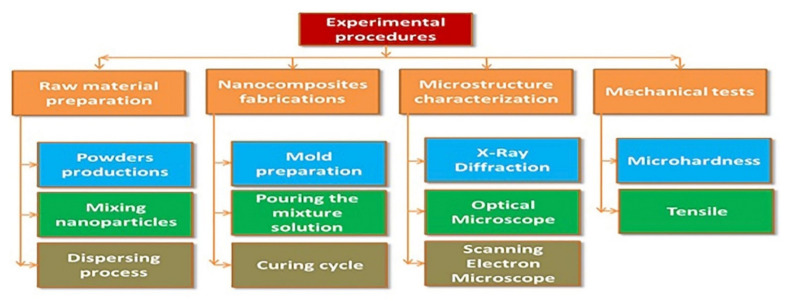
Flowchart of the detailed research methodology for the present work.

**Figure 3 polymers-13-04116-f003:**
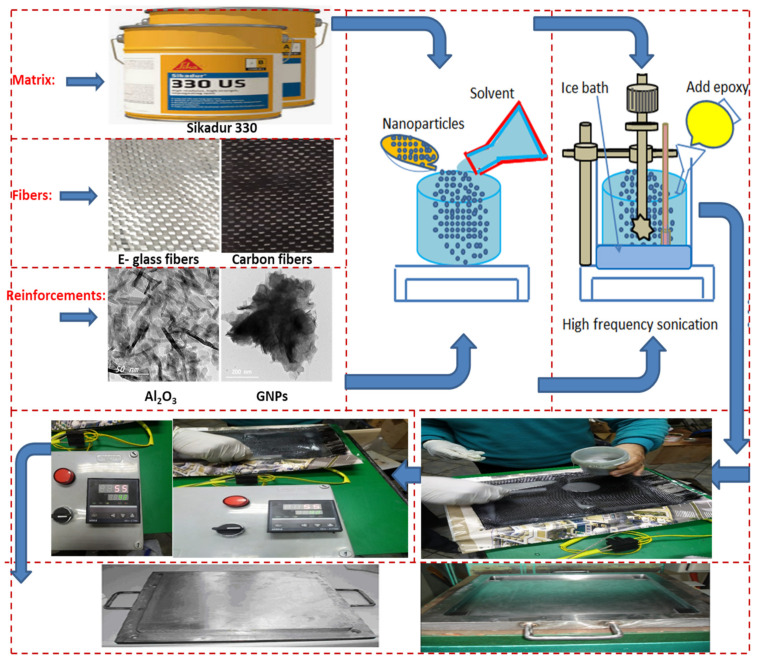
The procedures of dispersion, hand lay-up technique, and compression mold technology. Note: the images of Al_2_O_3_ and GNP nanoparticles were taken from reference [40].

**Figure 4 polymers-13-04116-f004:**
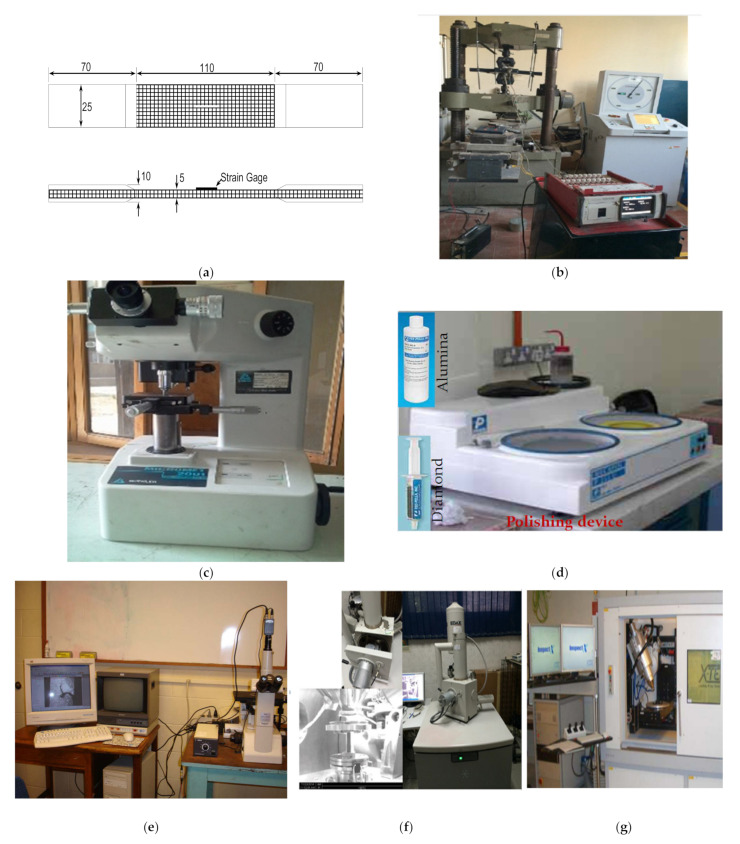
The procedures of microstructural analyses and mechanical tests: (**a**) dimensions of tensile test specimen, (**b**) universal test machine, (**c**) hardness test, (**d**) polishing machine, (**e**) optical microscope, (**f**) scanning electron microscope, and (**g**) X-ray diffraction.

**Figure 5 polymers-13-04116-f005:**
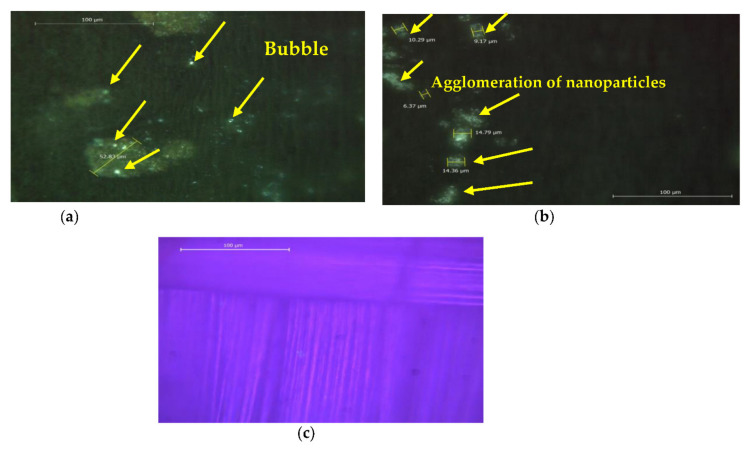
Optical microscope of prepared samples at: (**a**) S_1_, (**b**) S_2_, and (**c**) S_3_.

**Figure 6 polymers-13-04116-f006:**
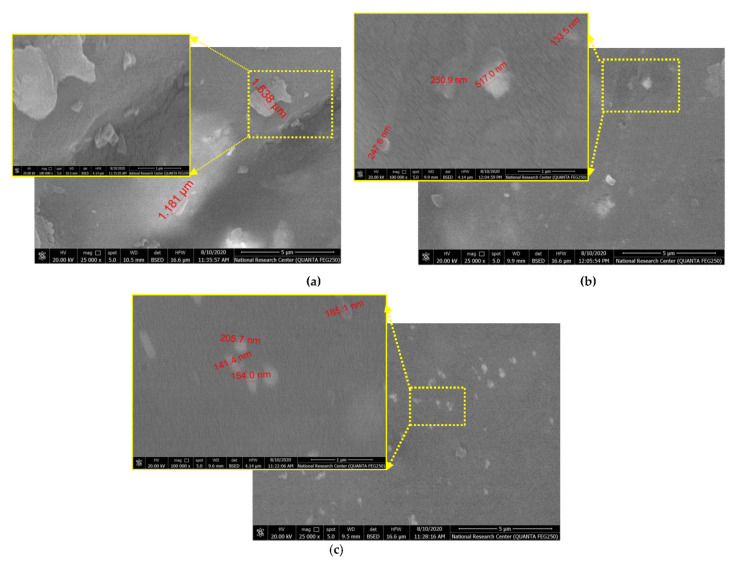
Scanning electron microscopy of prepared samples on top surface: (**a**) S_1_, (**b**) S_2_, and (**c**) S_3_.

**Figure 7 polymers-13-04116-f007:**
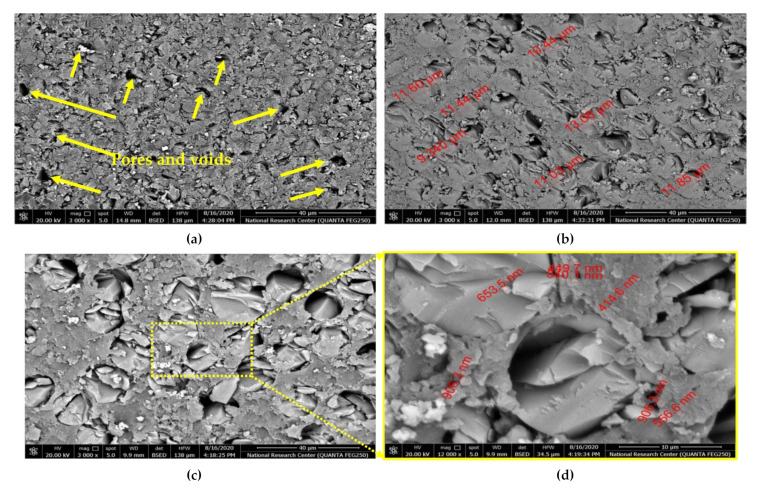
Scanning electron microscopy of prepared samples at cross-surface: (**a**) S_1_, (**b**) S_2_, (**c**) S_3_, and (**d**) high magnification scale of S_3_.

**Figure 8 polymers-13-04116-f008:**
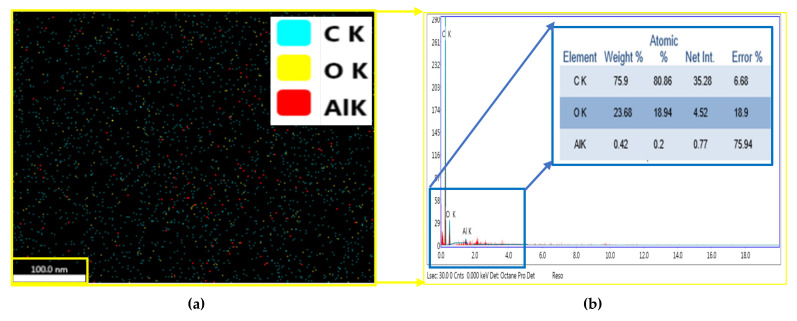
Map analysis in (**a**) and EDX in (**b**) by FE-SEM image for the prepared sample (S_2_), hybrid glass and carbon fibers with 1.5 wt.% GNPs and 1.5 wt.% Al_2_O_3_).

**Figure 9 polymers-13-04116-f009:**
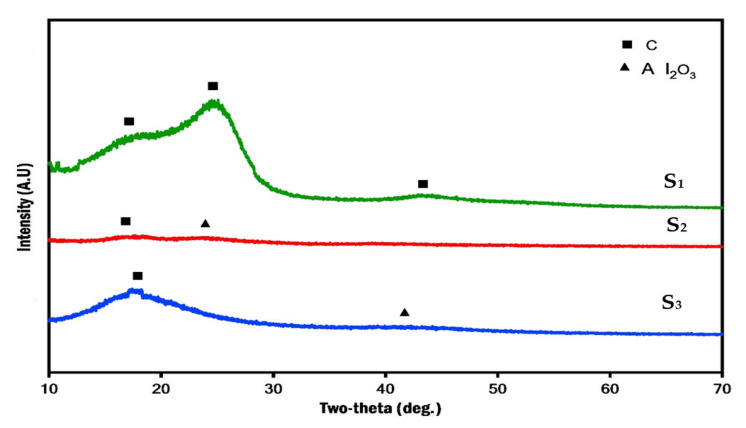
XRD patterns of prepared samples at: S_1_, S_2_ and S_3_.

**Figure 10 polymers-13-04116-f010:**
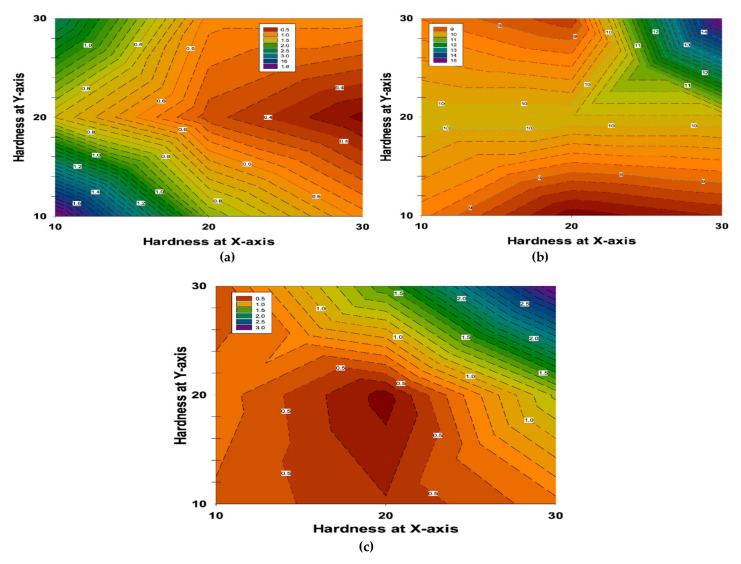
Hardness maps of fabricated specimens at: (**a**) S_1_, (**b**) S_2_, and (**c**) S_3_ on both the x and y-axes.

**Figure 11 polymers-13-04116-f011:**
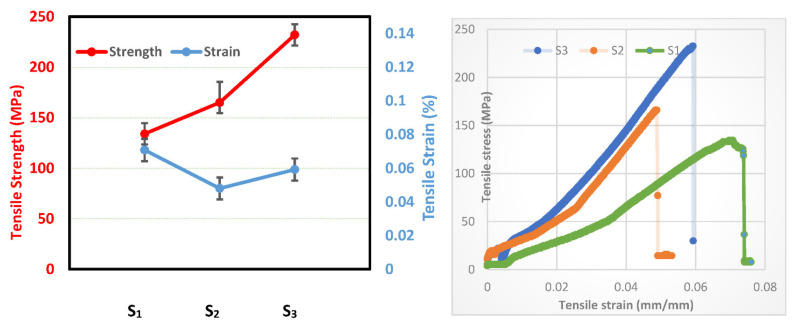
Tensile strength and strain of prepared samples at: S_1_, S_2_, and S_3_.

**Figure 12 polymers-13-04116-f012:**
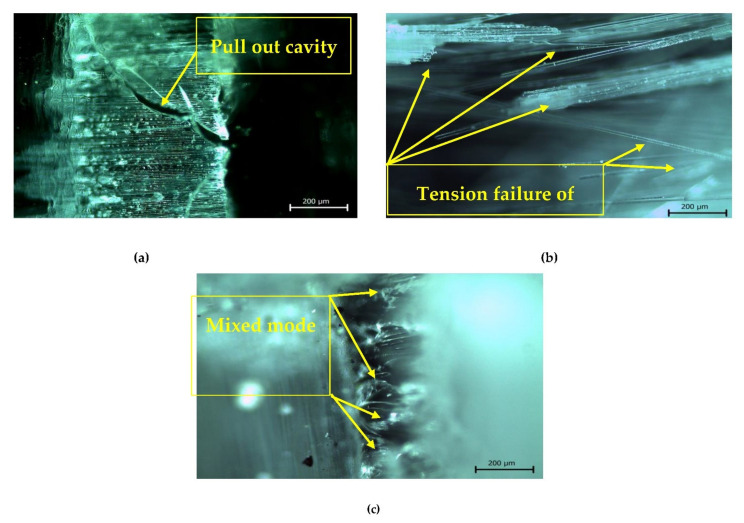
Fracture morphologies of prepared samples by optical microscope at: (**a**) S_1_, (**b**) S_2_, and (**c**) S_3_.

**Table 1 polymers-13-04116-t001:** Physical and mechanical properties of epoxy resin, E-glass fiber, carbon fiber, Al_2_O_3_, and GNPs.

Properties	Epoxy	E-Glass Fiber	Carbon Fiber	Al_2_O_3_	GNPs
Density (gram/cm^3^)	1.16	2.56	1.6	3.78	2.267
Tensile strength (MPa)	30	1400	2400	665	167
Tensile modulus (GPa)	4.1	72.3	228	210	2.4 × 10^3^
Poisson’s ratio	0.35	0.22	0.30	0.24	0.012

**Table 2 polymers-13-04116-t002:** Different types of composite structures used in the present experimental procedures.

Series	Specimen Name	Specimen Code	Reinforcement Type	Matrix Type	Ceramic Reinforcements
I	Hybrid glass and carbon fibers	S_1_	Carbon fibers + Glass fibers	Sikadur 330 epoxy	without
II	Hybrid glass and carbon fibers at (1.5% wt. GNPs + 1.5% wt. Al_2_O_3_)	S_2_	Carbon fibers + Glass fibers	Sikadur 330 epoxy	1.5% wt. GNPs + 1.5% wt. Al_2_O_3_
III	Hybrid glass and carbon fibers at 3% wt. Al_2_O_3_	S_3_	Carbon fibers + Glass fibers	Sikadur 330 epoxy	3%wt. Al_2_O_3_

## Data Availability

All the data generated during this study are included in this article.

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
