# Peer review of "Synthesis and Characterization of Hybrid Fiber-Reinforced Polymer by Adding Ceramic Nanoparticles for Aeronautical Structural Applications"

_polymers, 2021, doi:10.3390/polym13234116_

Round 1

Reviewer 1 Report

The presented manuscript lacks proper details of the material, testing and discussion. Reviewer suggests authors to revised the manuscript. Here are some points which needs to be take care of:

  1. Authors should mention the type of fibres they have used i.e. UD fabric or twill fabric?
  2. Authors should calculate the fibre volume fraction of each laminate before comparing the mechanical properties.
  3. Even while discussing the mechanical properties authors simply put tensile strength or elongation. They should mention its either UD strength or transverse?
  4. Authors should perform a detailed mechanical characterization including UD and transverse tensile and flexural, including DCB to check the influence of nanoparticles on fracture energy and ILSS as well.
  5. Authors mentioned the TEM analysis but reviewer didn't find any TEM images in the manuscript.

Author Response

-Dear editor:

Editor in Chief, Journal of Polymers

Thank you for giving me the opportunity to submit a revised draft of my manuscript titled " Synthesis and characterization of hybrid fibers reinforced polymer with adding ceramic nanoparticles for aeronautical structures applications" to Journal of Polymers. Manuscript ID: Polymers-1390876.

We appreciate the time and effort that you and the reviewers have dedicated to providing your valuable feedback on my manuscript. We are grateful to the reviewers for their insightful comments on my paper. We have been able to incorporate changes to reflect most of the suggestions provided by the reviewers. We have highlighted the changes within the manuscript.

Here is a point-by-point response to the reviewers’ comments and concerns.

* Reviewer 1:   

  • Comment 1: Authors should mention the type of fibers they have used i.e. UD fabric or twill fabric?
  • Response 1: Thank you for pointing this out. We agree with this comment, but in the experimental work: section “2. Materials and Methods” the type of fibers was mentioned in page 5 lines from 175 to 176. The kinds of used fibers in the experimental work were wove fibers. Moreover, the shape of fibers was elucidated in Fig.3.
  • Comment 2: Authors should calculate the fiber volume fraction of each laminate before comparing the mechanical properties.
  • Response 2: Thank you for pointing this out. We agree with this comment. The fiber volume fraction (νf) was measured experimentally according to ASTM D-3171-99. The average volume fraction was about at 64%.
  • Comment 3: Even while discussing the mechanical properties authors simply put tensile strength or elongation. They should mention its either UD strength or transverse?
  • Response 3: Although the reviewer’s comment is highly respected and such telling the UD strength or transverse would provide additional information, however, in the present study, the orientations of fibers were woven fabric in all directions as shown in Fig. 3. So, tensile test was one in any directions, this is because the orientations of fibers were woven as shown in Fig. 3.
  • Comment 4: Authors should perform a detailed mechanical characterization including UD and transverse tensile and flexural, including DCB to check the influence of nanoparticles on fracture energy and ILSS as well.
  • Response 4: Although the reviewer’s comment is highly respected, however, as mentioned in the previous comment, the orientations of fibers were woven fabric in all directions as shown in Fig. 3. The mechanical characterizations were performed with more detailed. As mentioned in the our manuscript “The composites specimens were cut in both longitudinal and transverse directions to ensure the isotropy for the fabricated composite”.
  • Comment 5: Authors mentioned the TEM analysis, but reviewer didn't find any TEM images in the manuscript.
  • Response 5: Thank you for pointing this out. We agree with this comment. Thanks to the reviewer. Yes, we didn’t conduct microstructure characterization by TEM, thus mentioned the TEM analysis has been deleted. The TEM analysis was only mentioned at the nanoceramics in Fig.3 before pre-synthesis.

We really appreciate your time and the valuable comments the reviewers. We believe that the changes we have made in response to them have led to clear improvements such that the manuscript is hopefully now suitable for publication. We look forward to hearing from you soon.

Best regards

Yours sincerely

The authors

Thank you very much

Reviewer 2 Report

An interesting and significant study on the carbon/glass fiber reinforced polymer composite incorporated the ceramic nanoparticles was conducted. The microstructure, XRD patterns, hardness, and tensile properties of composites were systematically investigated. The maximum tensile strength of hybrid composite was achieved when the content of nano- Al2O3 was 3 wt. %. The work has significance to promote the development and application of carbon/glass fiber hybrid composite. To improve the quality of paper, the authors should consider the following specific comments, make major changes and supplement the necessary information and related explanations.

  1. In the abstract part, the distribution mechanism of nanoparticles in resin polymer and its improvement mechanism on the tensile strength of composite should be clarified and added. Furthermore, the hybrid mechanism of carbon fiber and glass fiber (such as hybrid mode, volume ratio etc.) should also be analyzed, which is meaningful to readers.
  2. In the keywords, “Thin-Walled Structures” seems to be ambiguous, and it should be replaced.
  3. Figure 1 is not clear and needs to be replaced with high-quality pictures.
  4. Line 91-104, the authors mentioned the advantages of carbon fiber and glass fiber in terms of performance and price. However, the mechanical properties and long-term properties carbon/glass fiber hybrid composite should be summarized in detail. It is suggested that the author add the summary on hybrid mechanism, hybrid mode of carbon fiber and glass fiber and the long-term evolution under the loading and environment. Please see the following typical work on carbon/glass fiber hybrid composites: Journal of Materials Research and Technology, 2021, 14:2812-2831. Composites Science and Technology, 2009, 69: 432-437. Composite Structures, 2020; 246: 112418. In addition, more research on “carbon/glass hybrid” should be collected to improve the significance of present introduction.
  5. Line 105-119, the authors analyzed the potential advantages of nanotechnology to prepare some new polymer composites. However, the effect of nanomaterials on the improvement mechanism of polymer have not been deeply summarized, for example, the physical and chemical interaction between nanoparticles and epoxy resin. It is suggested that the authors consider the above contents and supplement the relevant analysis and summary.
  6. In Table 1, the tensile strength of carbon fiber and glass fiber is relatively low. Is it the strength of monofilament or fiber fabric? What are the types and manufacturer of carbon fiber and glass fiber used by the authors? It is well known that the tensile strength of CFRP can reach 2400 MPa. Please add relevant supplementary instructions.
  7. In table 2, what is the basis to choose the mass ratio of aluminum oxide and GNPs? In general, the content of nanoparticles has always been a variable to study the effect to the resin. Please explain this.
  8. The lower picture of Figure 3 is unclear. It is recommended to replace it with a high-definition picture.
  9. What is the layer structure of carbon fiber and glass fiber? The authors should consider what kind of layer structure can play the best hybrid effect of carbon fiber and glass fiber.
  10. From Figure 4 to Figure 8, the authors used SEM, EDX and XRD to analyze the dispersion state of nanoparticles in epoxy resin. However, the chemical and physical interactions among carbon fiber, glass fiber, epoxy resin and nanoparticles have not been deeply revealed. This includes the hybrid effect of carbon fiber and glass fiber, the interaction effect of nanoparticles and epoxy, and the interfacial bongding between fiber and resin. Please further summarize the contents of the above three parts.
  11. In Figure 10, the standard deviation of tensile data should be given.
  12. It is suggested to integrate the “discussion” in Part 4 into “results” in Part 3. At the same time, the author should summarize and analyze the mechanisms in question 10.

Author Response

-Dear editor:

Editor in Chief, Journal of Polymers

Thank you for giving me the opportunity to submit a revised draft of my manuscript titled " Synthesis and characterization of hybrid fibers reinforced polymer with adding ceramic nanoparticles for aeronautical structures applications" to Journal of Polymers. Manuscript ID: Polymers-1390876.

We appreciate the time and effort that you and the reviewers have dedicated to providing your valuable feedback on my manuscript. We are grateful to the reviewers for their insightful comments on my paper. We have been able to incorporate changes to reflect most of the suggestions provided by the reviewers. We have highlighted the changes within the manuscript.

Here is a point-by-point response to the reviewers’ comments and concerns.

* Reviewer 2:   

  • Comment 1: In the abstract part, the distribution mechanism of nanoparticles in resin polymer and its improvement mechanism on the tensile strength of composite should be clarified and added. Furthermore, the hybrid mechanism of carbon fiber and glass fiber (such as hybrid mode, volume ratio etc.) should also be analyzed, which is meaningful to readers.
  • Response 1: Thank you for pointing this out. We agree with this comment. The authors added the suggested corrections. In the abstract page 1 lines 22 and 23, the authors added that: In order to obtain good dispersion and distribution of additives nanoparticles inside resin matrix the ultrasonication process was used. In the abstract page 1 lines 29 and 30, the authors added that: Specifically, the tensile strength was improved from 133 MPa at the unreinforced specimen to reach 230 MPa at the reinforced specimen with 3 wt. % Al2O3. In the abstract page 1 lines from 32 till 34, the authors added that: The hybrid mode mechanism depends on the interaction among the mechanical properties of fiber, physical and chemical evolution of resin, bonding properties of fiber/resin interface and service environment. So, the hybrid mode of woven carbon and glass fibers at volume fraction 64% with the additive nanoparticles of GNPs / Al2O3 inside the resin was appropriate to produce aeronautical structures with extraordinary properties.
  • Comment 2: In the keywords, “Thin-Walled Structures” seems to be ambiguous, and it should be replaced.
  • Response 2: Thank you for pointing this out. The keyword “Thin-Walled Structures” was replaced to another word related to the application for the fabricated composite “Aeronautical structures applications”.
  • Comment 3: Figure 1 is not clear and needs to be replaced with high-quality pictures.
  • Response 3: Thank you for pointing this out. We agree with this comment. Fig.1 has been improved to a high-resolution quality image.
  • Comment 4: Line 91-104, the authors mentioned the advantages of carbon fiber and glass fiber in terms of performance and price. However, the mechanical properties and long-term properties carbon/glass fiber hybrid composite should be summarized in detail. It is suggested that the author add the summary on hybrid mechanism, hybrid mode of carbon fiber and glass fiber and the long-term evolution under the loading and environment. Please see the following typical work on carbon/glass fiber hybrid composites: Journal of Materials Research and Technology, 2021, 14:2812-2831. Composites Science and Technology, 2009, 69: 432-437. Composite Structures, 2020; 246: 112418. In addition, more research on “carbon/glass hybrid” should be collected to improve the significance of present introduction.
  • Response 4: Thanks to the reviewer for pointing this out. Anew TEXT has been added about hybrid mechanism, hybrid mode of carbon fiber and glass fiber and the long-term evolution under the loading and environment. “Currently, the hybrid fiber reinforced polymer (HFRP) composites proposed to use because have many benefits. Some of these advantages as to balance the material cost, performance and give high mechanical properties and corrosion resistance as well as fracture toughness and its resistance to the environmental effects [29-31]”. The authors added the suggested articles from reviewers to improve the significance of present introduction.

[29] Xian, G., Guo, R., Li, C., & Hong, B. Effects of rod size and fiber hybrid mode on the interface shear strength of carbon/glass fiber composite rods exposed to freezing-thawing and outdoor environments. JMR&T, 2021, 14, 2812-2831.

[30] Tsai, Y., Bosze, J., Barjasteh, E., Nutt, S. Influence of hygrothermal environment on thermal and mechanical properties of carbon fiber/fiberglass hybrid composites. Compos Sci Technol., 2009, 69 (4-3), 432-437.

[31] Li, C., Yin, X., Wang Y., Zhang L., Zhang Z., Liu, Y., Xian, G. Mechanical property evolution and service life prediction of pultruded carbon/glass hybrid rod exposed in harsh oil-well condition. Compos. Struct., 2020, 246, 112418.

  • Comment 5: Line 105-119, the authors analyzed the potential advantages of nanotechnology to prepare some new polymer composites. However, the effect of nanomaterials on the improvement mechanism of polymer have not been deeply summarized, for example, the physical and chemical interaction between nanoparticles and epoxy resin. It is suggested that the authors consider the above contents and supplement the relevant analysis and summary.
  • Response 5: Thank you for pointing this out. We agree with this comment. The authors added with briefing about the effect of nanomaterials on the improvement mechanism of polymer.

The authors added these statements:

“Several factors affecting on improving the mechanism of nanomaterials of PMCs, especially the physical and chemical interaction between nanoparticles and epoxy resin. Some of these factors related to the physical and chemical properties are concentration, and orientation as well as dispersion method, treatment state and manufacturing method. All these factors are related to both of fibers fabric, matrices and fillers. ”. 

The effect of nanomaterials on the improvement mechanism of polymer is depending through many factors such as the interaction among the mechanical properties of fiber, nanoparticle, physical and chemical evolution of resin, bonding properties of fiber/resin as well as nanoparticle interface and service environment [36].

  • Comment 6: In Table 1, the tensile strength of carbon fiber and glass fiber is relatively low. Is it the strength of monofilament or fiber fabric? What are the types and manufacturer of carbon fiber and glass fiber used by the authors? It is well known that the tensile strength of CFRP can reach 2400 MPa. Please add relevant supplementary instructions.
  • Response 6: Thanks to the reviewer. A new TEXT has been added about the types and manufacturer of carbon fiber and glass fiber used in our manuscript.

 The woven carbon fibers and glass fibers were supplied from Arab World for Financial Investments Company, Cairo, Egypt. On the other hand, two kinds of ceramic nanoparticles such as: aluminum oxide (Al2O3) and graphene nanoplatelets (GNPs) were used as strengthening nanoparticles in the resin matrix. The Sikadur 330 epoxy resin matrix was used to bond the four layers. Sikadur 330 epoxy resin was purchased from Sika Corporation. The kind of carbon and glass fibers were woven fibers fabric as supplied in the from of data sheet according to the production company. For tensile modulus of carbon fiber, the range obtained is between 1400 to 4800 MPa. The literature reports that the strength of carbon fiber, however, varies over a very wide range (1400 to 4800 MPa or more) according to, its production, under controlled conditions which should ensure a consistent strength value. The relevant supplementary instructions were added.

  • Comment 7: In table 2, what is the basis to choose the mass ratio of aluminum oxide and GNPs? In general, the content of nanoparticles has always been a variable to study the effect to the resin. Please explain this.
  • Response 7: Thanks to the reviewer.

The basis we stand on for selecting the mass ratio of aluminum oxide and GNPs was according to several literature survey, which were discussed the optimum mass ratio of nanoparticles inside the hybrid carbon and glass fiber reinforced polymers. Some of these articles were mentioned in the references of the current manuscript.

  • Comment 8: The lower picture of Figure 3 is unclear. It is recommended to replace it with a high-definition picture.
  • Response 8: Thank you for pointing this out. We agree with this comment. The lower picture of Figure 3 was replaced to be a high-definition picture.
  • Comment 9: What is the layer structure of carbon fiber and glass fiber? The authors should consider what kind of layer structure can play the best hybrid effect of carbon fiber and glass fiber.
  • Response 9: Thanks to the reviewer.

A new TEXT has been added about the layer structure of carbon fiber and glass fiber used in our manuscript.

The authors added this statement: “The kind of layer structure of fibers fabric, matrix, and fillers (nanoparticles) are played a critical role on scattering, distributing, and also adhesion state. Specifically, the atomic structure of the carbon fibers and glass fibers are like the graphite one, which is composed by flat sheets of carbon atoms (Graphene) placed following a regular hexagonal pattern. The difference between each of them is the way that the sheets are linked. The intermolecular strengths between each sheet are relatively smaller, (Van Der Waals), giving the graphite its soft and brittle properties. On the other hand, the structure of Al2O3 nanoparticles is fine spherical particle shape, while the structure of GNPs is fine flake shape. Therefore, the differences between structure shapes of these additive materials provide more benefits. Some of these benefits are: (i) helping to combine the matrix materials with fibers fabric, (ii) reinforcing the constituents together, and (iii) eliminating any bubbles, voids, and porosities in the microstructure. With another words, the combination between the fiber fabrics, matrix and nanoparticles has a significant benefit towards developing unique mechanical, chemical, and physical properties.”.”.

  • Comment 10: From Figure 4 to Figure 8, the authors used SEM, EDX and XRD to analyze the dispersion state of nanoparticles in epoxy resin. However, the chemical and physical interactions among carbon fiber, glass fiber, epoxy resin and nanoparticles have not been deeply revealed. This includes the hybrid effect of carbon fiber and glass fiber, the interaction effect of nanoparticles and epoxy, and the interfacial bonding between fiber and resin. Please further summarize the contents of the above three parts.
  • Response 10: Thanks to the reviewer.

For clarifying the contents of the above mentioned three parts, a new TEXT has been added as follows:

 In page 9-10

The authors added these statements:

The carbon atoms are bonded together in microscopic crystals that are mostly aligned parallel to the long axis of the fiber. This alignment makes the fiber show high tensile properties. Fillers are used in polymers for a variety of reasons, namely, to reduce cost, improve processing, control density, thermal conductivity, thermal expansion, electrical properties, magnetic properties, flame retardance, and to improve mechanical properties. Each filler type has different properties depending on particle size, shape, and surface chemistry. In general, the fillers can change the performance of polymer composites by changing the color, viscosity, barrier properties, curing rate, electrical and thermal properties, surface finish, shrinkage, etc. In general, the unique combination of the nanomaterial’s characteristics, such as size, mechanical properties, and low concentration is necessary to modify the polymer matrix properties have generated much interest in the field of nanocomposites. The properties of combination of the nanomaterial’s depend on several factors such as type of nanoparticle, surface treatments, polymer matrix, synthesis methods, and polymer nanocomposites morphology.

And also in page 8:

Generally, the surface area for a given volume is an important factor for nanoparticles. In another words, the geometrical shape of nanoparticles and their respective surface area to volume ratios are of more importance. This factor varies with the geometrical shape of nanoparticles, i.e. particle, layered, and fibrous materials. In the current study, we used three geometrical shapes of the constituents, the nanoparticles were particulate, while the fibers fabric were fibers and layers materials.

And also in page 9:

In general, the morphology of polymer composites can be divided into three categories, namely phase separated (microcomposite), intercalated nanocomposites, and exfoliated nanocomposites. In the phase separated (microcomposite), the clay nanoplatelets keep their crystal structure and the particle is in the microscale. In an intercalated (nanocomposite) stage, few polymer molecules penetrate into the fiber layers, with fixed interlayer spacing. In the exfoliated (nanocomposite) stage, the nanolayers are delaminated and individually dispersed in the continuous polymer matrix. The most desired structure for a nanoplatelet/polymer nanocomposite is for the nanofiller to be in the exfoliated state, as this provides maximum interfacial contact and best dispersion resulting in optimum nanocomposite performance.

  • Comment 11: In Figure 10, the standard deviation of tensile data should be given.
  • Response 11: Thanks to the reviewer. The standard deviation of tensile data has been added in Figure 10.
  • Comment 12: It is suggested to integrate the “discussion” in Part 4 into “results” in Part 3. At the same time, the author should summarize and analyze the mechanisms in question 10.
  • Response 12: Thanks to the reviewer. Although the reviewer’s comment is highly respected, however, one of the requirements of the polymers journal is to separate the section “results” from the section “discussion”.

We really appreciate your time and the valuable comments the reviewers. We believe that the changes we have made in response to them have led to clear improvements such that the manuscript is hopefully now suitable for publication. We look forward to hearing from you soon.

Best regards

Yours sincerely

The authors

Thank you very much

Reviewer 3 Report

This paper focuses on the synthesis and characterization of hybrid fibers reinforced polymer with adding ceramic nanoparticles for aeronautical structures purposes. Two different ceramic nanoparticles as alumina (Al2O3) and graphene nanoplatelets (GNPs) were chosen to incorporate into resin matrix. Please find below some major concerns regarding this manuscript.

  1. One main criticism I have of the work relates to the lack of details about the composite material. The preparation procedure is only introduced in Fig. 3. Then, paper “jump” to microstructural observations in Fig. 4, 5, 6 and 7. It is hard to follow what do the appearance of the composite material look like, where is the “cross surface”, how to obtain the microscopic image. Thus, it affects the overall understanding of materials research.

  1. The author introduced the microstructural observation results, and tried to analyze the reason in “Discussion” section. However, the explanation still lacks proofs and conviction. For example, Lines 506-525. Is the difference in the cross surface of the test piece related to the manufacturing process? Is it possible to add similar research literature to support it?

  1. In line 434 and 435 “The tensile strength and strain of fabricated specimens for S1, S2, and S3 were shown in Fig. 10. It is clear that the yield tensile strength of the samples is improved by adding the ceramic particles to the resin matrix.” The explanation does not appear sensical. In general, the strength of woven composite is determined by the reinforcement fabric.

  1. The paper studies the mechanical effects of reinforced particles, why not choose pure resin matrix as the research object, but instead choosing fabric reinforced composite material? The weave microstructure and micro-deformation effect enhances the complexity, which does not consider in the study.

  1. The mechanical properties are introduced in section “3.2”. The details of the samples, testing method, setup and images should be provided. Fig. 10 presents the result curves, each material S1, S2 and S3 has only been tested once, can't guarantee credible results. and the number of testing samples require be increased.

As there is too much work needed on the presentation, regret to advise that the paper is not suitable for publication in its present form.

Author Response

-Dear editor:

Editor in Chief, Journal of Polymers

Thank you for giving me the opportunity to submit a revised draft of my manuscript titled " Synthesis and characterization of hybrid fibers reinforced polymer with adding ceramic nanoparticles for aeronautical structures applications" to Journal of Polymers. Manuscript ID: Polymers-1390876.

We appreciate the time and effort that you and the reviewers have dedicated to providing your valuable feedback on my manuscript. We are grateful to the reviewers for their insightful comments on my paper. We have been able to incorporate changes to reflect most of the suggestions provided by the reviewers. We have highlighted the changes within the manuscript.

Here is a point-by-point response to the reviewers’ comments and concerns.

* Reviewer 3:

  • Comment 1: One main criticism I have of the work relates to the lack of details about the composite material. The preparation procedure is only introduced in Fig. 3. Then, paper “jump” to microstructural observations in Fig. 4, 5, 6 and 7. It is hard to follow what do the appearance of the composite material look like, where is the “cross surface”, how to obtain the microscopic image. Thus, it affects the overall understanding of materials research.
  • Response 1: Thanks to the reviewer for pointing this out. The authors corrected the details about the composite material.

The authors added a new TEXTS about the details about the composite material:

After specifying and preparing of the raw materials, we can be introduced the details about the synthesis of the hybrid composite structures. Three different kinds of aeronautical structures were selected in this study: (i) hybrid glass and carbon fibers without ceramic nanoparticles, (ii) hybrid glass and carbon fibers with equally percentage of 1.5 wt. % from Al2O3 and GNPs ceramic nanoparticles, and (iii) hybrid glass and carbon fibers with 3 wt. % Al2O3, see Table 2. GNPs and Al2O3 nanoparticles with various weight per-centages were added to Sikadur 330 epoxy resin and dispersed via simultaneous sonication and magnetic stirring, as can see Fig. 3. The dispersing process was carried out by sonication process through 0.5 cycles per second with 70% amplitude for 3 hours. The operation specifications of Hielscher ultrasonic processor were UP 200 S with 200 W and frequency 24 kHz. Suspensions of the GNPs and Al2O3 in the Sikadur 330 epoxy resin were prepared with the addition of dispersing agents as shown in Fig. 3. The concentration of GNPs and Al2O3 in the Sikadur 330 epoxy resin was varied according to Table 2. The ratio of hardener to epoxy resin was 1:2. Following, the dispersion process, the suspensions were exposed to centrifugation at 150 g for 3 hr. After preparing the mixture of ceramic nanoparticles and epoxy resin, we can be introduced the build of hybrid aeronautical structures. The construction of hybrid structures in this work were built with four layers which are consisting of two layers of glass fiber and two layers of carbon fiber. To construct the components of aeronautical structures by using hybrid FRPs with reinforcing ceramic nanoparticles, two techniques were implemented, in this work. The first technique was hand lay-up, and the second was compression-molding technology. The hand lay-up technique was implemented first to construct the part of the aeronautical structures. The first step of the hand lay-up technique was placed the fabrics woven fibers in-side the smooth surface of the mold with an insulator in between plastic sheet as release agent, after the mixture solution for the various composites was poured into the molds and covered with plastic transparent sheet as a release agent. In this moment, the ceramic nanoparticles were prepared via treating the surface through stearic acid as a non-reactive modifier. This step is to assist good adhesion be-tween ceramic nanoparticles and epoxy resin. The stearic acid was added to the solution of ethyl-acetate with stirring for half hour by an electric mixer at a velocity of 900 rpm. Then the ceramic nanoparticles were added to the mixture with stirring for another half hour, then the process of washing the nanoparticles by the solution of ethyl-acetate and filtrate until the excess stearic acid was removed. After that, carefully adding epoxy with nanoparticles with intermittent stirring for 30 minutes at 500 rpm. Then, add the mixture to the first layer of woven fabrics, till making sure that the first layer of woven fabrics is saturated with epoxy. After that, the second layer of woven fabrics is added sequentially. Then the process continued until the completion of the number of layers required. The cycle of the curing was done at 55°C for 30 min during saturation. After that, the second technique (compression mold technique) was implemented through preparing the mold with a flat and good surface finishing, as well as covering its internal dimensions (300×200 mm) were covered with a transparent plastic sheet as a release agent, as shown in Fig. 3. The fiber volume fraction (νf) was measured experimentally according to ASTM D-3171-99. The average volume fraction was about at 64%.

  • Comment 2: The author introduced the microstructural observation results, and tried to analyze the reason in “Discussion” section. However, the explanation still lacks proofs and conviction. For example, Lines 506-525. Is the difference in the cross surface of the test piece related to the manufacturing process? Is it possible to add similar research literature to support it?
  • Response 2: Thanks to the reviewer for pointing this out. The authors added similar research literature to support it.

The authors added these statements:

The hybrid mode mechanism depends on the interaction among the mechanical properties of fiber, physical and chemical evolution of resin, bonding properties of fiber/resin interface and service environment.

Several factors affecting on improving the mechanism of nanomaterials of PMCs, especially the physical and chemical interaction between nanoparticles and epoxy resin. Some of these factors related to the physical and chemical properties are concentration, and orientation as well as dispersion method, treatment state and manufacturing method. All these factors are related to both of fibers fabric, matrices, and fillers. Moreover, the effect of nanomaterials on the improvement mechanism of polymer is depending through many factors such as the interaction among the mechanical properties of fiber, nanoparticle, physical and chemical evolution of resin, bonding properties of fiber/resin as well as nanoparticle interface and service environment [36].

  • Comment 3: In line 434 and 435 “The tensile strength and strain of fabricated specimens for S1, S2, and S3 were shown in Fig. 10. It is clear that the yield tensile strength of the samples is improved by adding the ceramic particles to the resin matrix.” The explanation does not appear sensical. In general, the strength of woven composite is determined by the reinforcement fabric.
  • Response 3: Thanks to the reviewer for pointing this out. The authors corrected this point:

“It is clear that the tensile strength of the samples is improved by adding the ceramic particles to the resin matrix”. The explanation was added in section discussion.

A new TEXT has been added for more clarification:

The tensile strength of the fiber-reinforced composites mainly relies on various factors namely fiber orientation, length of fiber, strength, fiber content, fillers, bonding between fiber and matrix and weave style [48, 49]. By adding the nanoparticles of alumina (Al2O3) and graphene nanoplatelets (GNPs) to the resin matrix, the yield tensile strength of the samples is improved, since the nanoparticles act as impediments to the movement of interferences within the base material, which reduces the possibility of plastic deformation and the density of the resin impact on the mechanical properties [50]. A similar behavior was observed in [44] in which the addition of CNT–Al2O3 improved the mechanical properties of the fibrous composites.

[44]        Li, W., et al., On improvement of mechanical and thermo-mechanical properties of glass fabric/epoxy composites by incorporating CNT–Al2O3 hybrids. Composites science and technology, 2014. 103: p. 36-43.

[48]        Shibata, S., Y. Cao, and I. Fukumoto, Press forming of short natural fiber-reinforced biodegradable resin: Effects of fiber volume and length on flexural properties. Polymer testing, 2005. 24(8): p. 1005-1011.

[49]        Gowda, T.M., A. Naidu, and R. Chhaya, Some mechanical properties of untreated jute fabric-reinforced polyester composites. Composites Part A: applied science and manufacturing, 1999. 30(3): p. 277-284.

[50]        Mahan, H.M., D. Mahjoob, and N.A. Rashid, Influence of Silica and Nano Alumina on Mechanical Properties of Glass Fiber Reinforced Epoxy Composite Systems. Journal of Advanced Research in Dynamical and Control Systems, 2018. 10(2): p. 888-895.

  • Comment 4: The paper studies the mechanical effects of reinforced particles, why not choose pure resin matrix as the research object, but instead choosing fabric reinforced composite material? The weave microstructure and micro-deformation effect enhances the complexity, which does not consider in the study.
  • Response 4: Thanks to the reviewer.

According to your highly respect comment, it can be easily compared the mechanical properties of produced specimens with the epoxy resin matrix through available data in table 1; where the table 1 shows the physical and mechanical properties of epoxy resin matrix and different types of fibers fabrics. On the other hand, in the current work, this study focused the effect of adding ceramic nanoparticles on the mechanical properties. So, the reference case was selected as hybrid carbon and glass fibers without any additions of ceramic nanoparticles. Finally, the main effect was adding nanoparticles on hybrid fibers fabric with sonication technique. 

  • Comment 5: The mechanical properties are introduced in section “3.2”. The details of the samples, testing method, setup and images should be provided. Fig. 10 presents the result curves, each material S1, S2 and S3 has only been tested once, can't guarantee credible results. and the number of testing samples require be increased.
  • Response 5: Thanks to the reviewer. A new TEXTS have been added about the major concerns regarding.

The details of the samples, testing method, setup and images should be provided.

The details of the samples:

After specifying and preparing of the raw materials, we can be introduced the details about the synthesis the hybrid composite structures. Three different kinds of aeronautical structures were selected in this study: (i) hybrid glass and carbon fibers without ceramic nanoparticles, (ii) hybrid glass and carbon fibers with equally percentage of 1.5 wt. % from Al2O3 and GNPs ceramic nanoparticles, and (iii) hybrid glass and carbon fibers with 3 wt. % Al2O3, see Table 2. GNPs and Al2O3 nanoparticles with various weight percentages were added to Sikadur 330 epoxy resin and dispersed via simultaneous sonication and magnetic stirring, as can see Fig. 3. The dispersing process was carried out by sonication process through 0.5 cycles per second with 70% amplitude for 3 hours. The operation specifications of Hielscher ultrasonic processor were UP 200 S with 200 W and frequency 24 kHz. Suspensions of the GNPs and Al2O3 in the Sikadur 330 epoxy resin were prepared with the addition of dispersing agents as shown in Fig. 3. The concentration of GNPs and Al2O3 in the Sikadur 330 epoxy resin was varied according to Table 2. The ratio of harden-er to epoxy resin was 1:2. Following, the dispersion process, the suspensions were ex-posed to centrifugation at 150 g for 3 hr. After preparing the mixture of ceramic nanoparticles and epoxy resin, we can be introduced the build of hybrid aeronautical structures. The construction of hybrid structures in this work were built with four layers which are consisting of two layers of glass fiber and two layers of carbon fiber. To construct the components of aeronautical structures by using hybrid FRPs with reinforcing ceramic nanoparticles, two techniques were implemented, in this work. The first technique was hand lay-up, and the second was compression-molding technology. The hand lay-up technique was implemented first to construct the part of the aeronautical structures. The first step of the hand lay-up technique was placed the fabrics woven fibers inside the smooth surface of the mold with an insulator in between plastic sheet as release agent, after the mixture solution for the various composites was poured into the molds and covered with plastic transparent sheet as a release agent.

In this moment, the ceramic nanoparticles were prepared via treating the surface through stearic acid as a non-reactive modifier. This step is to assist good adhesion between ceramic nanoparticles and epoxy resin. The stearic acid was added to the solution of ethyl-acetate with stirring for half hour by an electric mixer at a velocity of 900 rpm. Then the ceramic nanoparticles were added to the mixture with stirring for another half hour, then the process of washing the nanoparticles by the solution of ethyl-acetate and filtrate until the excess stearic acid was removed. After that, carefully adding epoxy with nanoparticles with intermittent stirring for 30 minutes at 500 rpm. Then, add the mixture to the first layer of woven fabrics, till making sure that the first layer of woven fabrics is saturated with epoxy. After that, the second layer of woven fabrics is added sequentially. Then the process continued until the completion of the number of layers required. The cycle of the curing was done at 55°C for 30 min during saturation. After that, the second technique (compression mold technique) was implemented through preparing the mold with a flat and good surface finishing, as well as covering its internal dimensions (300×200 mm) were covered with a transparent plastic sheet as a release agent, as shown in Fig. 3. The fiber volume fraction (νf) was measured experimentally according to ASTM D-3171-99. The average volume fraction was about at 64%.

The testing method:

After presenting the details about the synthesis of the hybrid composite structures, we can be introduced the testing methods. The microstructural observations were analyzed through optical microscopy (OM) and scanning electron microscope (SEM) and X-ray diffraction (XRD) as well as Energy Dispersive X-ray spectroscopy (EDS) and the mechanical testing were evaluated by using tensile and hardness tests.

The composites specimens were cut in both longitudinal and transverse directions to ensure the isotropy for the fabricated composite. In order to prepare the morphology of the fabricated composites specimens, the specimens were mounted, then mechanically ground, and polished according to the standard metallography practices. The phases present in the fabricated composites were identified by X-ray diffraction (XRD) technique using Cu K-alpha radiation (λ = 1.541 Å). XRD scans were carried out with a step size of 0.02â—¦ and a long scan range (2θ) of 5-60Ëš. Tensile test according to ASTM standard was performed at room temperature with a strain rate of 5 mm/min as shown in Fig. 4. Hard-ness test was also carried out at room temperature using Shimadzu Vickers microhard-ness testing by applying 200 g load and 10 sec dwell time on the fabricated nanocomposites samples.

The setup and images should be provided: the setup and images were provided.

Fig. 10 presents the result curves, each material S1, S2 and S3 has only been tested once, can't guarantee credible results. “The standard deviation of tensile data has been added in Figure 10”.

The number of testing samples require be increased: Thirty-six specimens were evaluated in this work, where three samples in the one case were evaluated in all tests, then average value was introduced”.

We really appreciate your time and the valuable comments the reviewers. We believe that the changes we have made in response to them have led to clear improvements such that the manuscript is hopefully now suitable for publication. We look forward to hearing from you soon.

Best regards

Yours sincerely

The authors

Thank you very much

Reviewer 4 Report

Dear Editor and Authors,

I read carefully the article entitled ‘Synthesis and characterization of hybrid fibers reinforced polymer with adding ceramic nanoparticles for aeronautical structures applications’’. I can conclude that is an interesting paper, but some issues must be clarified:

  1. page 6 row 226- 150 g??? please modify
  2. In Figure 5, sample S1 has a different scale bar. Please modify.
  3. Please add in the caption of Figure 7, the description of Figure 7d.
  4. In Figure 8a, the percentage of Al is 0???? Please modify
  5. Please explain Figure 9 more adequately.
  6. In Figure 10, is the hardness higher in sample S1 than in sample S3? Please explain.

Best regards,

Author Response

Review 4:

I read carefully the article entitled ‘Synthesis and characterization of hybrid fibers reinforced polymer with adding ceramic nanoparticles for aeronautical structures applications’’. I can conclude that is an interesting paper, but some issues must be clarified:

  1. Comment: page 6 row 226- 150 g??? please modify.

Response: Thanks to the reviewer for pointing this out, the sentence was modified.

  1. Comment: In Figure 5, sample S1 has a different scale bar. Please modify.

Response: Thanks to the reviewer for pointing this out, the figure was modified.

  1. Comment: Please add in the caption of Figure 7, the description of Figure 7d.

Response: Thanks to the reviewer for pointing this out, the caption of figure 7 was modified, the description of figure 7d was added.

  1. Comment: In Figure 8a, the percentage of Al is 0???? Please modify

Response: Thanks to the reviewer for pointing this out, the percentage of Al was modified.

  1. Comment: Please explain Figure 9 more adequately.

Response: Thanks to the reviewer for pointing this out, figure 9 was explained with more adequately.

  1. Comment: In Figure 10, is the hardness higher in sample S1 than in sample S3? Please explain.

Response: Thanks to the reviewer for pointing this out, this is because the hardness value in sample S1 was measured in the fiber’s fabrics, while the hardness value in sample S2 was measured in the polymer resin epoxy. 

Thank you very much

Reviewer 5 Report

The manuscript is interesting but need further improvements to become publishable:

  1. All figures need improvement in terms of clarity and quality:
    • Figure 1 distorted
    • Figure 2 lower part distorted
    • Fig 4 (b)(c) distorted
    • Fig 6 (b)(c) cannot be read
    • Figure 8 change scale, I cannot understand anything
    • Fig 9 make black and white compatible
    • Fig 11 b/w compatible
  2. In several sections, English language needs improvements
  3. In introduction state the innovation of your research and clear research scope
  4. Describe all parameters of your experimental work, the reader should be reproduce
  5. In conclusions add a short description of our work. Also, please add useful conclusions for industry and researchers (potential future research). Furthermore add quantitative findings in conclusions.

Author Response

Review 5:

The manuscript is interesting but need further improvements to become publishable:

Comment: All figures need improvement in terms of clarity and quality:

    • Figure 1 distorted
    • Figure 2 lower part distorted
    • Fig 4 (b)(c) distorted
    • Fig 6 (b)(c) cannot be read
    • Figure 8 change scale, I cannot understand anything
    • Fig 9 make black and white compatible
    • Fig 11 b/w compatible

Response: Thanks to the reviewer for pointing this out, all figures have been improved to be cleared.

Comment: In several sections, English language needs improvements.

Response: Thanks to the reviewer for pointing this out, English language have been improved to be cleared.

Comment: In introduction state the innovation of your research and clear research scope.

Response: Thanks to the reviewer for pointing this out, the innovation and scope of the current paper have been mentioned.

So, the main scope of the present article is the production of aeronautical structures material reinforced with GNPs and Al2O3 nanoparticles which with far lighter weight and higher strength than those produced by the fiber-reinforced Polymers (FRPs) only. However, an attempt was made for the first time to produce aeronautical structures material reinforced with GNPs and Al2O3 nanoparticles through scattering the nanoparticles by high-frequency sonication technique then using hand-lay up and compression molding technique to produce thin-walled structures for aeronautical purposes.

Comment: Describe all parameters of your experimental work, the reader should be reproduce.

Response: Thanks to the reviewer for pointing this out, the parameters of the experimental work were mentioned.

To investigate the effect of adding ceramic nanoparticles on microstructural and mechanical characteristics of hybrid carbon and glass fibers reinforced polymers, three different kinds of aeronautical structures are selected in this study as: (i) hybrid glass and carbon fibers without ceramic nanoparticles, (ii) hybrid glass and carbon fibers with equally percentage of 1.5 wt. % from Al2O3 and GNPs ceramic nanoparticles, and (iii) hybrid glass and carbon fibers with 3 wt. % Al2O3, see Table 2. Accordingly, the main parameter of this work is to determine the optimum weight percent of ceramic nanoparticles inside hybrid glass and carbon fibers for aeronautical purposes.

Comment: In conclusions add a short description of our work. Also, please add useful conclusions for industry and researchers (potential future research). Furthermore add quantitative findings in conclusions.

Response: Thanks to the reviewer for pointing this out, the conclusions have been modified, short description was added, and useful conclusions for industry and researchers were added as well as quantitative findings in conclusions were added.

Thank you very much

Round 2

Reviewer 1 Report

Paper can be accepted.

Author Response

Review 1:

Comment: Paper can be accepted.

Response: thank you very much for reviewing.

Reviewer 2 Report

The authors have replied carefully to the reviewers' comments, and the quality of the paper has been greatly improved. Therefore, I suggest that the paper can be accepted in present form.

Author Response

Review 2:

Comment: The authors have replied carefully to the reviewers' comments, and the quality of the paper has been greatly improved. Therefore, I suggest that the paper can be accepted in present form.

Response: thank you very much for reviewing.

Reviewer 3 Report

Although the authors have improved the manuscript, there is still something unclear to me. Please find below some major concerns regarding this manuscript.

  1. The main criticism is that the paper is not well targeted. The last review comment points out that it is lack of details about the composite material. The author tries to supplement some test instruments figures, but the content of the paper still jumps directly from material fabrication to micro-level analysis, lack of meso-level and macro-level description and figures. Although much information about the preparation process is added, still hard to follow what do the appearance of the composite material look like. It affects the overall understanding of materials research. In addition, the distortion appears in the figure 3 and figure 4.

  1. In line 588-691, the authors try to connect the cross surface of pores, voids to the uniform dispersing and mechanical properties in materials. However, I struggled to understand the connection between the newly added text to the target. “Moreover, the effect of nanomaterials on the improvement mechanism of polymer is depending through many factors such as the interaction among the mechanical properties of fiber, nanoparticle, physical and chemical evolution of resin, bonding properties of fiber/resin as well as nanoparticle interface and service environment [36].”

  1. “The tensile strength of the fiber-reinforced composites mainly relies on various factors namely fiber orientation, length of fiber, strength, fiber content, fillers, bonding between fiber and matrix and weave style”. As for fabric reinforced composite, reference [44] is conducted on the flexural strength, not the tensile strength as in the paper. The explanation still does not appear sensical. Should the tensile strain-stress curves be provided?

As there is too much work needed on the presentation, regret to advise that the paper is not suitable for publication in its present form.

Author Response

Review 3:

Although the authors have improved the manuscript, there is still something unclear to me. Please find below some major concerns regarding this manuscript.

  1. Comment: The main criticism is that the paper is not well targeted. The last review comment points out that it is lack of details about the composite material. The author tries to supplement some test instruments figures, but the content of the paper still jumps directly from material fabrication to micro-level analysis, lack of meso-level and macro-level description and figures. Although much information about the preparation process is added, still hard to follow what do the appearance of the composite material look like. It affects the overall understanding of materials research. In addition, the distortion appears in the figure 3 and figure 4.

Response: Thanks to the reviewer for pointing this out, the details about the composite material were elucidated.

  1. Comment: In line 588-691, the authors try to connect the cross surface of pores, voids to the uniform dispersing and mechanical properties in materials. However, I struggled to understand the connection between the newly added text to the target. “Moreover, the effect of nanomaterials on the improvement mechanism of polymer is depending through many factors such as the interaction among the mechanical properties of fiber, nanoparticle, physical and chemical evolution of resin, bonding properties of fiber/resin as well as nanoparticle interface and service environment [36].”

Response: Thanks to the reviewer for pointing this out, the sentences were modified.

  1. Comment: “The tensile strength of the fiber-reinforced composites mainly relies on various factors namely fiber orientation, length of fiber, strength, fiber content, fillers, bonding between fiber and matrix and weave style”. As for fabric reinforced composite, reference [44] is conducted on the flexural strength, not the tensile strength as in the paper. The explanation still does not appear sensical. Should the tensile strain-stress curves be provided?

Response: Thanks to the reviewer for pointing this out, the sentence was modified;  the tensile strain-stress curves was elucidated.

Thanks to reviewing, thank you very much 

Reviewer 4 Report

I recommend the publication of the article.

Author Response

Thank you for reviewing, thank you very much.

Reviewer 5 Report

In first round I made the bellow comments:

The manuscript is interesting and well presented. It needs extension on the test methodology (even references to proper test standards), the reader should be able to reproduce. Furthermore, the discussion on test results and analysis should be extended and improved providing more information. Please add quantitative results in your conclusions.

English language is generally ok but it can be improved in some phrases and terms. For example the "salt environment" could be written somehow as Cl- attack.

I understand, due to a bad communication with editorial office, the authors did not received them. The manuscript has been improved after revision, so the authors should consider previous comments only.

Author Response

Thank you for reviewing.

Test methodology was provided.

The discussion on test results and analysis were improved.

English language was improved.

Thank you very much.

Round 3

Reviewer 3 Report

The authors have tried to improve the manuscript on hybrid fibers reinforced polymer with adding ceramic nanoparticles. In my opinion, the previous problems still exist, and not sufficiently complete to warrant publication. Regret to advise that the paper is not suitable for publication in its present form.

Author Response

All suggested comments have been modified. Thank you to review.

 Thank you very much.